# Comparative Transcriptome Analysis Reveals the Transcriptional Alterations in Growth- and Development-Related Genes in Sweet Potato Plants Infected and Non-Infected by SPFMV, SPV2, and SPVG

**DOI:** 10.3390/ijms20051012

**Published:** 2019-02-26

**Authors:** Jiang Shi, Lin Zhao, Baiyuan Yan, Yueqing Zhu, Huasheng Ma, Wenyue Chen, Songlin Ruan

**Affiliations:** 1Institute of Crop Science, Hangzhou Academy of Agricultural Sciences, Hangzhou 310024, China; tomatoman@126.com (J.S.); 15925638398@163.com (L.Z.); hzhsma@163.com (H.M.); 2Jiande Seed Management Station, Hangzhou 311600, China; zjjdyby@126.com; 3Linan District Forestry and Agriculture Bureau, Hangzhou 311300, China; 20050020@zafu.edu.cn; 4Laboratory of Plant Molecular Biology & Proteomics, Institute of Biotechnology, Hangzhou Academy of Agricultural Sciences, Hangzhou 310024, China

**Keywords:** sweet potato plants infected by SPFMV, SPV2 and SPVG, sweet potato plants non-infected by SPFMV, SPV2 and SPVG, co-infection, transcriptome profiling, gene ontology, pathway analysis

## Abstract

Field co-infection of multiple viruses results in considerable losses in the yield and quality of storage roots in sweet potato. However, little is known about the molecular mechanisms underlying developmental disorders of sweet potato subjected to co-infection by multiple viruses. Here, a comparative transcriptomic analysis was performed to reveal the transcriptional alterations in sweet potato plants infected (VCSP) and non-infected (VFSP) by *Sweet potato mild mottle virus* (SPFMV), *Sweet potato virus Y* (SPV2) and *Sweet potato virus G* (SPVG). A total of 1580 and 12,566 differentially expressed genes (DEGs) were identified in leaves and storage roots of VFSP and VCSP plants, respectively. In leaves, 707 upregulated and 773 downregulated genes were identified, whereas 5653 upregulated and 6913 downregulated genes were identified in storage roots. Gene Ontology (GO) classification and pathway enrichment analysis showed that the expression of genes involved in chloroplast and photosynthesis and brassinosteroid (BR) biosynthesis in leaves and the vitamin biosynthetic process in storage roots was inhibited by co-infection of three viruses: SPFMV, SPV2, and SPVG. This was likely closely related to better photosynthesis and higher contents of Vitamin C (Vc) in storage roots of VFSP than that of VCSP. While some genes involved in ribosome and secondary metabolite-related pathways in leaves and alanine, aspartate, and glutamate metabolism in storage roots displayed higher expression in VCSP than in VFSP. Quantitative real-time PCR analysis demonstrated that the expression patterns of 26 DEGs, including 16 upregulated genes and 10 downregulated genes were consistent with the RNA-seq data from VFSP and VCSP. Taken together, this study integrates the results of morphology, physiology, and comparative transcriptome analyses in leaves and storage roots of VCSP and VFSP to reveal transcriptional alterations in growth- and development-related genes, providing new insight into the molecular mechanisms underlying developmental disorders of sweet potato subjected to co-infection by multiple viruses.

## 1. Introduction

Sweet potato (*Ipomoea batatas*. Lam), in the family Convolvulaceae, is a light-loving and short-day crop. The main producing areas of sweet potato in the world are located at 40 degrees north latitude. Asia has the largest area of cultivated sweet potato, followed by Africa, and North America is the third largest. China is the largest sweet potato-producing country in the world, with a total planting area of over 3 million hectares and an annual output of 0.72 billion tons [1]. After being infected by viruses, sweet potato gradually shows degeneration phenomena such as declines in yield and quality and loses its desirable characteristics. According to a survey in certain areas in China, including Shandong, Anhui, Beijing, Jiangsu, and other provinces, the yield loss of sweet potato caused by virus diseases can exceed 50%, resulting in an economic loss of approximately five hundred million dollars [2]. Therefore, the adverse effect of virus diseases on sweet potato production is of increasing concern.

Because sweet potato is mainly planted by cuttings, many viruses can spread through insects and sap, and the co-infection rate is high. For example, *Sweet potato feather mottle virus* (SPFMV), *Sweet potato virus G* and *Sweet potato virus C* belonging to *Potato virus Y* showed co-infection in the field [3]. The sweet potato virus disease (SPVD) caused by the synergistic infection of *Sweet potato chlorosis dwarf virus* (SPCDV) and SPFMV is a destructive disease of sweet potato [4,5], which can cause losses of 70%–100% [6]. Effects on yields caused by SPFMV or *Sweet potato chlorotic stunt virus* (SPCSV) alone were minor, but co-infection by two or more viruses caused 50% losses in yield [7]. In Uganda, mixed infections of four viruses, including *Sweet potato chlorotic spot virus* (SPCFV), SPCSV, SPFMV, and *Sweet potato mottle mosaic virus* (SPMMV), were found in wild species and cultivated sweet potato [8]. Jiang et al. [9] detected five viruses in 24 samples from Shandong Province, including 23 co-infection samples that were classified into 11 categories of infection. Based on the observation that the tip of the sweet potato stem often has little or no virus, virus-free sweet potato can be obtained via tissue culture technology, which can effectively restore the desirable characteristics of the varieties, improve the yield and quality of the storage roots, and prolong the shelf life of the varieties [10]. Virus-free meristems were grown into plants, which were kept under insect-proof conditions and away from other sweet potato material for distribution to farmers after another cycle of reproduction [7].

The physiological and biochemical mechanisms underlying the improved yield and quality of virus-free sweet potato storage roots, including increased net photosynthetic rate and chlorophyll content, higher activities of antioxidant enzymes (superoxide dismutase (SOD), peroxidase (POD), and catalase (CAT)) and reduced lipid peroxidation levels, have widely been reported [11,12,13]. However, the molecular mechanisms underlying the immune response of plants to the virus mainly involve vsiRNA. There are mainly two mechanisms of vsiRNA immunity to virus. vsiRNAs can be recruited by Argonaut (AGO )proteins and directly degrade virus RNA through post-transcriptional gene silencing (PTGS) [14]. vsiRNAs can also bind to AGO proteins and act on virus DNA and silence virus genes by enhancing the methylation of DNA [15,16].

Transcriptomics is a powerful tool for discovering differentially expressed genes and has been widely applied in many crop species [17]. Recently, transcriptome sequencing analysis has been performed in purple sweet potato [18] and a sweet potato progenitor [19]. However, little is known about differential alterations in transcriptome profiles between sweet potato plants infected (VCSP) and non-infected (VFSP). Here, a comparative transcriptomic analysis was performed to reveal the significantly upregulated and downregulated genes. Gene Ontology (GO) classification of the proteins encoded by these genes was used to analyze their cellular locations, molecular functions, and biological processes. A pathway analysis was performed to reveal the biological pathways involving these genes. This study may provide new insight into the transcriptional alterations in VFSP and VCSP.

## 2. Results

### 2.1. Phenotypes and Growth Indexes of VCSP and VFSP

To identify phenotypes of VCSP and VFSP, we first performed the detection of viruses in their leaves and storage roots using Q-PCR. The results showed that three viruses, including SPFMV, SPV2, and SPVG, were found in VCSP leaves and storage roots but not in VFSP leaves and storage roots (Figure 1C). VCSP leaves exhibited a small chlorotic and mosaic phenotype, whereas VFSP leaves displayed a normal green phenotype (Figure 1A). Interestingly, fluorescence imaging analysis showed that VFSP leaves exhibited strong blue fluorescence, whereas VCSP leaves displayed inhomogeneous fluorescence with photobleaching (Figure 1B). The values of leaf length (LL), leaf width (LW), and Fv/Fm, in VFSP leaves were significantly higher than those in VCSP leaves (Figure 1D,F). Similarly, the contents of total chlorophyll, chlorophyll a, and chlorophyll b were significantly higher in VFSP than in VCSP (Figure 1E). In addition, the yield and contents of Vc and beta-carotene of storage roots in VFSP were significantly higher than those in VCSP (Figure 2B–D), whereas the content of starch in storage roots was lower in VFSP than in VCSP (Figure 2A).

### 2.2. Gene Expression Profiles of Leaves and Storage Roots in VCSP and VFSP

To understand the molecular mechanisms underlying developmental disorders of sweet potato subjected to co-infection by multiple viruses, comparative transcriptomic analysis was performed to discover differentially expressed genes in leaves and storage roots of VFSP and VCSP. As shown in Figure 3, a total of 1580 and 12,566 DEGs were identified in leaves and storage roots, respectively. Interestingly, in both leaves and storage roots, the number of downregulated genes was higher than that of upregulated genes. In leaves, 707 upregulated and 773 downregulated genes were identified, whereas 5653 upregulated and 6913 downregulated genes were identified in storage roots.

### 2.3. GO Classification of Differential Expression Genes

To find out the correlation between phenotypic differences and gene expression, we then performed a GO classification of 707 and 5653 upregulated genes in leaves and storage roots, respectively. The results showed that the proteins encoded by these genes in leaves were significantly assigned to 52 cellular components (Appendix A). Of these, the top five cellular components were photosystem (GO:0009521), photosystem I (GO:0009522), chloroplast thylakoid (GO:0009534), thylakoid (GO:0009579), and plastid thylakoid (GO:0031976) (Figure 4A), which was closely associated with chloroplast. Subsequently, proteins encoded by the upregulated genes were classified into 39 functional categories (Appendix A). The top five categories were chlorophyll binding (GO:0016168), tetrapyrrole binding (GO:0046906), carbon-oxygen lyase activity acting on polysaccharides (GO:0016837), pectate lyase activity (GO:0030570), and catalytic activity (GO:0003824) (Figure 4A). Finally, these categories were considered to be mainly involved in 92 biological processes (Appendix A). Of these processes, the top five were photosynthesis (GO:0015979), protein–chromophore linkage (GO:0018298), carbohydrate biosynthetic process (GO:0016051), single-organism carbohydrate metabolic process (GO:0044723), and polysaccharide biosynthetic process (GO:0000271) (Figure 4A). These processes were closely related to photosynthesis, which was consistent with the result of phenotypic differences in leaf color, chlorophyll content, and fluorescence characteristics between VFSP and VCSP.

Similarly, we performed a GO classification of 5653 upregulated genes in storage roots and discovered that the proteins encoded by these genes were significantly assigned to three cellular components (Appendix A), including anchored component of membrane (GO:0031225), nucleolus (GO:0005730), and endoplasmic reticulum lumen (GO:0005788) (Figure 4B). Next, proteins encoded by the upregulated genes were classified into 12 functional categories (Appendix A), Of these, the top five functional categories were four iron, four sulfur cluster binding (GO:0051539), oxidoreductase activity acting on other nitrogenous compounds as donors (GO:0016661), iron–sulfur cluster binding (GO:0051536), metal cluster binding (GO:0051540) and xyloglucan:xyloglucosyl transferase activity (GO:0016762) (Figure 4B). Finally, these proteins were assigned to be involved in 56 biological processes (Appendix A). Of them, the top three biological processes were sulfur compound metabolic process (GO:0006790), sulfur compound biosynthetic process (GO:0044272), and organonitrogen compound biosynthetic process (GO:1901566) (Figure 4B). Interestingly, there were three genes assigned to be involved in vitamin biosynthetic process, which might be responsible for that VFSP had higher content of vitamin C than VCSP.

We also carried out a GO classification of 773 and 6913 downregulated genes in leaves and storage roots, respectively. The results indicated that the proteins encoded by these genes in leaves were significantly assigned to 27 cellular components (Appendix A). Of these components, the top five were ribosome (GO:0005840), intracellular ribonucleoprotein complex (GO:0030529), ribonucleoprotein complex (GO:1990904), mitochondrial part (GO:0044429), and mitochondrial envelope (GO:0005740) (Figure 4C). Next, proteins encoded by the downregulated genes were classified into 25 functional categories (Appendix A). Of these categories, the top five were chitin binding (GO:0008061), chitinase activity (GO:0004568), threonine-type endopeptidase activity (GO:0004298), threonine-type peptidase activity (GO:0070003), and ammonia-lyase activity (GO:0016841) (Figure 4C). Finally, they were assigned to be involved in 155 biological processes (Appendix A). Of these processes, the top five were regulation of RNA metabolic process (GO:0051252), regulation of transcription, DNA-templated (GO:0006355), regulation of nucleic acid-templated transcription (GO:1903506), regulation of RNA biosynthetic process (GO:2001141) and transcription (Figure 5). Similarly, proteins encoded by these genes in storage roots were significantly assigned to one cell component, extracellular region (GO: 0005576) (Figure 4D). Secondly, proteins encoded by the down-regulated genes were classified into 23 functional categories (Appendix A). Of these categories, the top five were hydrolase activity, hydrolyzing O-glycosyl compounds (GO:0004553), aldo-keto reductase (NADP) activity (GO:0004033), hydrolase activity, acting on glycosyl bonds (GO:0016798), glutamate-ammonia ligase activity (GO:0004356) and ammonia ligase activity (GO:0016211) (Figure 4D). At last, they were assigned to be involved in 39 biological processes (Appendix A). Of these processes, the top five were polysaccharide catabolic process (GO:0000272), pectin catabolic process (GO:0045490), macromolecule catabolic process (GO:0009057), nitrogen fixation (GO:0009399), and beta-glucan catabolic process (GO:0051275) (Figure 4D).

### 2.4. Pathway Analysis of Differentially Expressed Genes

To determine the involvement of these differentially expressed genes in leaves and storage roots, we performed a pathway analysis to identify the potential target genes. The upregulated genes in leaves were identified to be involved in 10 distinct metabolic pathways. Of them, the top five were photosynthesis—antenna proteins (ko00196), photosynthesis (ko00195), porphyrin and chlorophyll metabolism (ko00860), carbon fixation in photosynthetic organisms (ko00710), and nitrogen metabolism (ko00910) (Figure 6A, Appendix A). Similarly, the upregulated genes in storage roots were identified to be involved in 19 distinct metabolic pathways. Of them, the top five were linoleic acid metabolism (ko00591), thiamine metabolism (ko00730), monobactam biosynthesis (ko00261), selenocompound metabolism (ko00450), and sulfur metabolism (ko00920) (Figure 6B, Appendix A).

The downregulated genes in leaves were identified to be involved in 15 distinct metabolic pathways. Of them, the top five pathways were ribosome (ko03010), flavonoid biosynthesis (ko00941), phenylalanine metabolism (ko00360), stilbenoid, diarylheptanoid and gingerol biosynthesis (ko00945), and biosynthesis of amino acids (ko01230) (Figure 6C, Appendix A). Similarly, the downregulated genes in storage roots were identified to be involved in 19 distinct metabolic pathways. The top five pathways were alanine, aspartate and glutamate metabolism (ko00250); monoterpenoid biosynthesis (ko00902); galactose metabolism (ko00052); GABAergic synapse (ko04727); and two-component system (ko02020) (Figure 6D, Appendix A).

### 2.5. Validation of differentially expressed candidate genes

To validate the Illumina sequencing data and the expression patterns of the DEGs revealed by RNA-Seq, qRT-PCR was performed to examine the expression patterns of 26 DEGs, including 16 upregulated genes and 10 downregulated genes (Figure 4). qRT-PCR results showed that 10 genes in leaves and 6 genes in storage roots involved in photosynthesis in VFSP showed higher abundance than those in VCSP: antenna proteins, photosynthesis, porphyrin and chlorophyll metabolism, carbon fixation in photosynthetic organisms, steroid biosynthesis, linoleic acid metabolism, thiamine metabolism and monobactam biosynthesis, including (Figure 5A,C). Furthermore, 4 genes in leaves and 6 genes in storage roots in VFSP showed lower abundance than those in VCSP, including genes involved in the ribosome, flavonoid biosynthesis, alanine, aspartate and glutamate metabolism, monoterpenoid biosynthesis and galactose metabolism (Figure 5B,D), which was consistent with the RNA-seq data from VFSP and VCSP.

## 3. Discussion

Field co-infection of multiple viruses in sweet potato is a common phenomenon worldwide, which has resulted in the great losses of the yield and quality of storage roots in sweet potato [6,7,8]. Here, VCSP leaves and storage roots was examined through Q-PCR analysis with primers of five viruses including SPFMW, SPV2, SPVG, SPCSV and SPVC. Most of VCSP plants were co-infected by three viruses including SPFMW, SPV2 and SPVG, and the other two viruses were not found in all virus-checked VCSP plants, whereas no virus was detected in virus-checked VFSP plants. In VCSP plants, the typical phenotypes with small chlorotic and mosaic leaves were observed and the yield and quality of storage roots were dramatically reduced. As we expected, photosynthetic characteristics including the chlorophyll content and maximal photochemical efficiency (Fv/Fm) of leaves in VCSP were significantly lower than that in VFSP, indicating that the photosystem of VCSP leaves was obviously impaired by the virus infection, leading to reduced photosynthesis and yield in VCSP. To explain the differences between VCSP and VFSP at the molecular level, comparative transcriptome analysis was performed. Interestingly, the number of upregulated genes in storage roots was much higher than that in leaves. We speculate that storage roots may be in an important stage of reserve substance synthesis, while these viruses may also play a role in reduced the expression of a large number of related genes to reserve substance synthesis in VCSP just at this time. This is partially confirmed by the results of subsequent GO and Kyoto Encyclopedia of Genes and Genomes (KEGG) analysis. GO classification and KEGG pathway analysis showed that the majority of proteins encoded by upregulated genes in VFSP leaves were localized in chloroplast and its structural components, including photosystem I, photosystem II, thylakoid and stroma, and involved in multiple biological processes associated with photosynthesis and porphyrin and chlorophyll metabolism. Obviously, expression of these chloroplast and photosynthesis genes is initially inhibited by co-infection with three viruses. In addition, we also found that genes involved in carbon fixation and nitrogen metabolism in VFSP plants were shown commonly upregulated expression pattern. It has been reported that carbon (C) and N metabolism are tightly coordinated in the fundamental processes that permit plant growth, for example photosynthesis and N uptake [20,21,22,23]. Therefore, we believe that expression of some genes involved in photosynthesis, carbon fixation, and nitrogen metabolism is negatively regulated by co-infection with three viruses, resulting in decreased growth and development of VCSP plants.

Similarly, certain upregulated genes involved in water-soluble vitamin metabolic process, particularly the water-soluble vitamin biosynthetic process in storage roots in VFSP, were found. Indeed, storage roots in VFSP showed a higher content of Vc than those in VCSP. Therefore, we speculate that an enhanced water-soluble vitamin biosynthetic process may be responsible for the higher accumulated contents of Vc in storage roots of VFSP. Conversely, co-infection of multiple viruses reduced the expression of genes involved in water-soluble vitamin biosynthetic process, which may be responsible for the higher accumulated contents of Vc in storage roots of VCSP. In addition, we found that there were two genes involved in other vitamin synthetic processes including thiamine biosynthetic process and thiamine-containing compound biosynthetic process. Therefore, we speculate that these viruses not only reduce Vc synthesis, but also may adversely affect the synthesis of other vitamins in VCSP storage roots.

We found that 10 upregulated genes in VFSP leaves were involved in steroid biosynthesis. Of these, five genes, including CAS1 (K01853), SMO1 (K14423), HYD1 (K01824), SMO2 (K14424), and STE1 (K00227), are involved in brassinosteroid (BR) biosynthesis. In storage roots, there is one gene involved in brassinosteroid (BR) biosynthesis. BRs play an essential role in diverse developmental programs, including cell expansion, vascular differentiation, etiolation, and reproductive development [24]. It has been demonstrated that BRs can induce resistance of tobacco, and rice to bacterial, fungal, and viral pathogens [25,26]. Recent evidence showed that *Cucumber mosaic virus* (CMV) resistance was positively regulated by BR levels, and BR signaling was required for this BR-induced CMV tolerance [27]. Furthermore, BR treatment alleviated photosystem damage, improved antioxidant enzyme activity and induced defense-associated gene expression under CMV stress in *Arabidopsis* [27]. Interestingly, although the combination of MeJA and BL treatment resulted in a significant reduction in *Rice black-streaked dwarf virus* (RBSDV) infection compared with a single BL treatment, MeJA application efficiently suppressed the expression of BR pathway genes, and this inhibition depended on the JA coreceptor OsCOI1, indicating that JA-mediated defense can suppress the BR-mediated susceptibility to RBSDV infection [26]. We speculate that BR biosynthesis in leaves of sweet potato may be adversely affected by co-infection with these viruses. In other words, BR biosynthesis may negatively be regulated by co-infection with these viruses, resulting in reduced growth and development of VCSP plants.

## 4. Materials and Methods

### 4.1. Plant Materials and Growth Conditions

Seedlings of sweet potato plants infected (VCSP) and non-infected (VFSP) with SPFMV, SPV2, and SPVG from the variety ‘Zheshu 6025′ were supplied by Hangzhou Academy of Agricultural Sciences (Hangzhou, Zhejiang Province, China). Two hundred of candidate VFSP seedlings were obtained through virus elimination by tissue culture of growth points of VCSP seedlings. Forty-eight non-infected seedlings were obtained through Q-PCR analysis of candidate VFSP seedlings with primers of five viruses including SPFMW, SPV2, SPVG, SPCSV and SPVC. Most of VCSP seedlings were co-infected with three viruses including SPFMW, SPV2 and SPVG, while these viruses were not found in all VCSP seedlings. Seedlings of VCSP and VFSP were planted in different pots in the same shed in mid-March. Before collection, each plant was detected by Q-PCR to ensure the sample consistency. Sweet potato storage roots were harvested at 150 d. The FW of VCSP and VFSP storage roots in three plots was calculated. The yield of each plot was calculated.

### 4.2. Measurement of Leaf Length (LL) and Width (LW)

The lengths and widths of 10 leaves from four replicates of VCSP and VFSP plants were measured. The standard errors (SE) of mean LL or LW were calculated.

### 4.3. Determination of Chlorophyll Contents

The contents of chlorophyll a and chlorophyll b were directly measured from the crude chlorophyll extracts of sweet potato leaves. A total of 0.2 g leaf tissues was homogenized in ethanol at 4 °C as described by Porra et al. [28]. The homogenates were centrifuged, and their fluorescence at 662, 645 and 470 nm was measured with an ultraviolet visible optical spectrometer UV2550 (manufactured by SHIMADZU Corporation, Kyoto, Japan).

### 4.4. Assay of Maximal Photochemical Efficiency (Fv/Fm)

Chlorophyll fluorescence was determined with a chlorophyll fluorescence imaging system (IMAGING PAM; Heinz Walz, Effeltrich, Germany). To measure the maximal quantum efficiency of PSII (Fv/Fm), sweet potato leaves were dark-adapted for 30 min. The measured light intensity for normal light and saturating light was 1 and 10, respectively. Fv/Fm was also measured with an FMS-2 pulse amplitude fluorimeter (Hansatech Instruments Ltd., Kings Lynn, Norfolk, UK). Minimal fluorescence (Fo) was measured under a weak pulse of modulating light over a 0.8-s period, and maximal fluorescence (Fm) was obtained after a saturating pulse of 0.7 s at 8000 μmol m ^−2^ s ^−1^.

### 4.5. Assay of Starch Content

Two to 5 g of the finely ground sample was weighed and filtered through 40 mesh or finer into in a funnel with slow filter paper. The fat in the sample was washed five times with 50 mL of petroleum ether, after which the petroleum ether was discarded. The residue was washed many times with 150 mL of ethanol (85% to volume ratio) to remove soluble sugars. The ethanol solution was drained, the residue in the funnel was rinsed with 100 mL of water, and the solution was transferred to a 250 mL cone bottle. Thirty milliliters of hydrochloric acid (1 + 1) was added, and a condensing tube was connected, after which the solution was held for 2 h in a boiling water bath. After the reflux was finished, the solution was immediately cooled. When the hydrolysate of the sample was cooled, 2 drops of methyl red indicator were added. First, sodium hydroxide solution (400 g/L) turned yellow and then hydrochloric acid (1 + 1) was added to the sample. The hydrolysate turned red. Then, 20 mL of lead acetate solution (200 g/L) was added and the solution was shaken well for 10 min. Twenty milliliters of sodium sulfate solution (100 g/L) was added to remove excessive lead. The solution and residue were shaken and added to a 500 mL volumetric flask, and the conical flask was washed with water. The solution was incorporated into the volumetric flask and diluted with water to scale. Twenty milliliters of the filtrate was retained for determination. A total of 5 mL of alkaline tartaric acid copper solution and 5 mL of alkaline tartaric acid copper solution were collected and placed in 150 mL cone bottles. Ten milliliters of water and two beads of glass beads were added, and 9 mL of a glucose standard solution from the burette was added. The control was heated to boiling and kept boiling for 2 min. The glucose was added continuously at a rate of one drop every two seconds until the blue solution faded. The total volume of the glucose standard solution was recorded, and three replicates were performed at the same time, and the average value was taken. The copper solution of each 10 mL (A and B 5 mL) of alkaline tartaric acid was equivalent to the mass of m1 (mg) of glucose.

### 4.6. HPLC Analysis of Vitamin C

Sweet potato storage roots (10–20 g) were homogenized with the same extraction solution (20 g/L HPO_3_) for 30 min. Mixture supernatants were then recovered by filtration and constituted the raw extracts. After the reduction of raw extracts, vitamin C was quantified by HPLC (Waters system) using an isocratic gradient equipped with a reversed-phase C 18 column (Waters, Spherisorb ODS 2) (5 µm packing) (250 × 4.6 mm id). Ascorbic acid was eluted under the following conditions: injected volume 20 µL; oven temperature 20 °C; mobile phase, A: 6.8 g/L KH_2_PO_4_ + 0.91 g/L HTAB (hexadecyl trimethyl ammonium bromide); B: 100% MeOH; solvent mixture: A/B (98/2) (*v*/*v*). The flow rate was 0.7 mL min^−1^, and the total elution time was 10 min. Detection was performed with an III-1311 Milton Roy fluorimeter (Ivyland, PA, USA) with a wavelength of 245 nm. Quantification was carried out by external calibration with ascorbic acid. The calibration curve was set from 0.5 to 50 µg/mL ascorbic acid.

### 4.7. Assay of Total Beta-Carotene

Sweet potato storage roots (1–5 g) were homogenized with 75 mL of the extraction solution (13.3 g/L ascorbic acid and ethanol), and the mixture was shaken at 60 °C for 30 min. Mixture supernatants were then recovered by filtration and constituted the raw extracts. After the saponification and extraction of raw extracts, beta-carotene was quantified by HPLC (Waters system) using an isocratic gradient equipped with a reversed-phase C_30_ column (Waters, Spherisorb ODS 2) (5 µm packing) (150 × 4.6 mm id). Beta-carotene was eluted under the following conditions: injected volume 20 µL; oven temperature 30 °C; mobile phase, A: MeOH, ACN, H_2_O (73.5/24.5/2, *v*/*v*/*v*); B: 100% MTBE (methyl tertbutyl ether). The flow rate was 1.0 mL·min^−1^, and the total elution time was 10 min. Detection was performed with an III-1311 Milton Roy fluorimeter (Ivyland, PA, USA) with a wavelength of 450 nm. Quantification was carried out by external calibration with beta-carotene. The calibration curve was set from 0.5 to 10 µg/mL beta-carotene.

### 4.8. RNA Sequencing and Data Analysis

Sweet potato leaves and storage roots from ten plants were pooled as an independent experimental replicate. Three independent experimental replicates were used for transcriptomic analysis. Total leaf RNA was isolated from sweet potato leaves and storage roots using TRIzol reagent (Invitrogen, Carlsbad, CA, USA) according to the manufacturer′s protocols, dissolved in RNase-free water and then used to construct the transcriptome sequence library using the NEBNext Ultra RNA Library Prep Kits for Illumina (NEB, Ipswich, MA, USA) following the manufacturer′s instructions. Index codes were added to attribute sequences to each sample. Finally, 125 bp paired-end reads were generated using Illumina HiSeq 2500 (Novogene, Beijing, China). Clean reads were obtained by removing the reads containing adapter or poly-N and the low-quality reads from raw data. The reads were aligned to the genome using the TopHat (2.0.9) software. To measure gene expression level, the total number of reads per kilobase per million reads (RPKM) of each gene was calculated based on the length of this gene and the counts of reads mapped to this gene. RPKM values were calculated based on all the uniquely mapped reads. The genes with RPKM ranging from 0 to 3 were considered at a low expression level, the genes with RPKM ranging from 3 to 15 at a medium expression level, and the genes with RPKM above 15 at a high expression level. Differential expression analysis was conducted using the DESeq R package (1.10.1). The resulting p values were adjusted using Benjamini and Hochberg’s approach for controlling the false discovery rate. Genes with an adjusted p value <0.05 identified by DESeq were assigned as differentially expressed. GO annotation was performed using the Blast2GO software (GO association was performed by a BLASTX against the NCBI NR database). GO enrichment analysis of differentially expressed genes was then performed with the BiNGO plugin for Cytoscape. Over-presented GO terms were identified using a hypergeometric test with the significance threshold of 0.05 after the Benjamini and Hochberg FDR correction. KEGG enrichment analysis of differentially expressed genes was performed using the KOBAS (2.0) [29] software.

### 4.9. Verification of RNA-Seq Data by Quantitative Real-Time PCR (qRT-PCR)

To test the reliability of RNA-seq data (Appendix A), a set of the top ten upregulated genes in three replicates was selected for qRT-PCR. Specific primers were designed with the Primer Express software (Applied Biosystems, Foster City, CA, USA) and synthesized by Sangon (Shanghai, China). cDNA was synthesized from 1 μg of total RNA using the PrimeScript RT reagent Kit (Takara, Dalian, China). Real-time RT-PCR was performed on the ABI 7500 Real-Time PCR System (Applied Biosystems) using the 2× SYBR green PCR master mix (Applied Biosystems). HongSu ARF (Genbank No. JX177359.1) was used as an internal standard. Three independent experimental replicates were analyzed for each sample, and data are indicated as the mean ± SE (*n* = 3). Twenty-nine pairs of primers were designed for specific transcript amplification of 3 virus genes and 26 differentially expressed genes (Appendix A).

### 4.10. Statistical Analysis

Three independent experimental replicates were analyzed for each treatment, and data are indicated as the mean ± SE (*n* = 3). Analyses of variance (ANOVA) were conducted by Duncan’s multiple range test. Before analysis of variance, percentages were transformed according to y = arcsin[sqr(x/100)]. All data were analyzed according to a factorial model and replicates as random effects. Means were compared among treatments by LSD (least significant difference) at 0.05 confidence level.

## 5. Conclusions

This study integrates the results of morphology, physiology, and comparative transcriptome analysis of the leaves and storage roots of VCSP and VFSP to address the differences in their underlying molecular mechanisms. Co-infection with three viruses, including SPFMV, SPV2, and SPVG, significantly reduces the expression of many genes involved in photosynthesis and photosynthesis-related pathways in VCSP, which adversely affects the development and growth of sweet potato through these pathways, resulting in a decrease in the yield and quality of storage roots in VCSP. Therefore, our findings provide new insight into transcriptional alterations in certain genes involved in photosynthesis, porphyrin, and chlorophyll metabolism, vitamin biosynthetic processes, BR biosynthesis, carbon fixation, and nitrogen metabolism in VCSP and VFSP, which helps to reveal the underlying molecular mechanism of developmental disorders of sweet potato subjected to co-infection by multiple viruses.

## Figures and Tables

**Figure 1 ijms-20-01012-f001:**
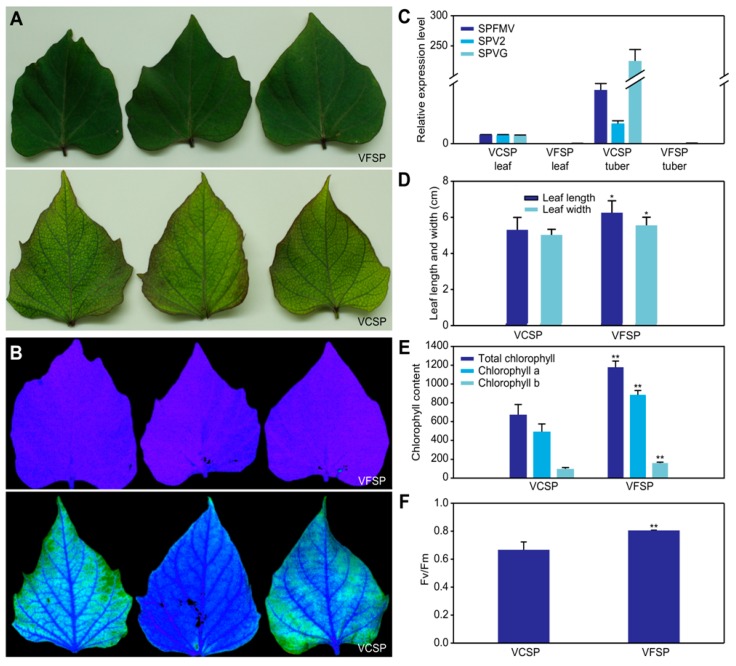
Phenotypes and growth indexes of sweet potato plants infected (VCSP) and non-infected (VFSP) leaves. (**A**) Phenotypes of VCSP and VFSP leaves. (**B**) Fluorescence images of VCSP and VFSP leaves. (**C**) Three virus genes were not detected in leaf and tuber of sweet potato. (**D**) Length and width of VCSP and VFSP leaves. (**E**) Content of chlorophyll a, b, and total chlorophyll. (**F**) Fv/Fm of VCSP and VFSP leaves. Three independent experimental replicates were analyzed for each treatment, and data are indicated as the mean ± SE (*n* = 3). Independent *t*-test was performed to check difference between VFSP and VCSP (** *p* < 0.01; * *p* < 0.05).

**Figure 2 ijms-20-01012-f002:**
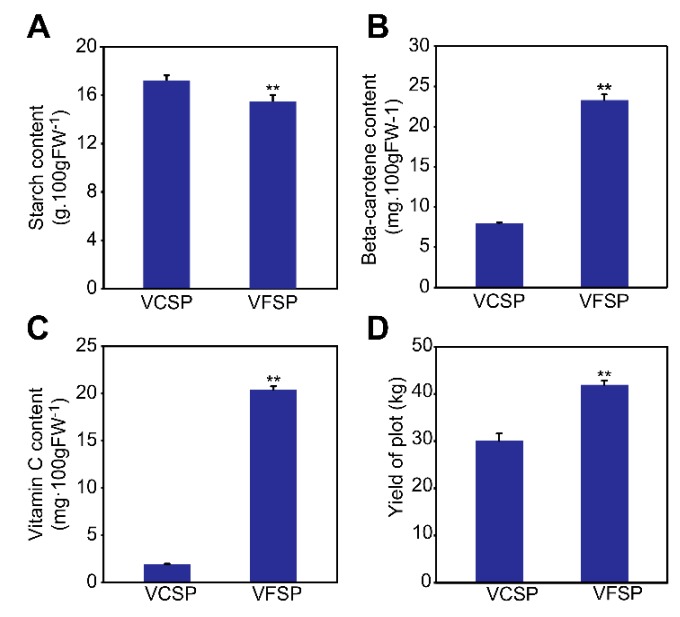
Comparison of yield and quality of VCSP and VFSP storage roots. (**A**) Content of starch. (**B**) Content of beta-carotene. (**C**) Content of vitamin C. (**D**) Yield of plot. Three independent experimental replicates were analyzed for each treatment, and data are indicated as the mean ± SE (*n* = 3). Independent *t*-test was performed to check difference between VFSP and VCSP (** *p* < 0.01; * *p* < 0.05).

**Figure 3 ijms-20-01012-f003:**
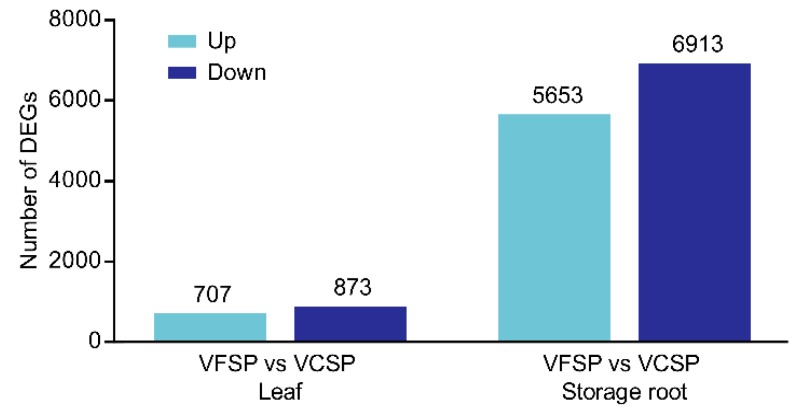
Gene expression profile of leaves and storage roots in VCSP and VFSP.

**Figure 4 ijms-20-01012-f004:**
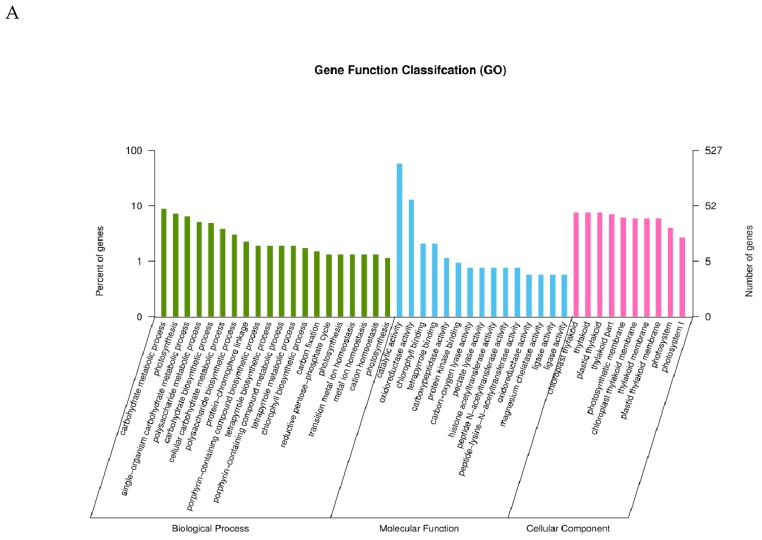
Gene Ontology (GO) classification of differentially expressed genes (DEGs) identified. GO classification of differentially up-regulated genes of VFSP/VCSP (**A**) leaves and (**B**) storage roots. GO classification of differentially down-regulated genes of VFSP/VCSP (**C**) leaves and (**D**) storage roots.

**Figure 5 ijms-20-01012-f005:**
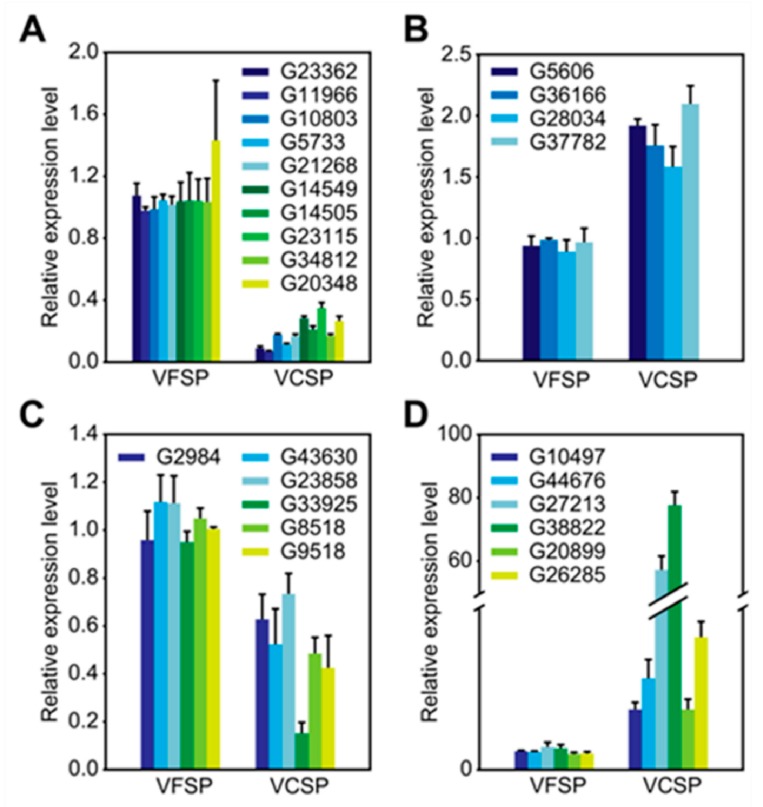
Validation of differentially expressed candidate genes. **A**,**B**: qRT-PCR analysis of ten upregulated genes (**A**) and four downregulated genes (**B**) in VFSP and VCSP leaves. **C**,**D**: qRT-PCR analysis of six upregulated genes (**C**) and six downregulated genes (**D**) in VFSP and VCSP storage roots. Three independent experimental replicates were analyzed for each sample, and data are indicated as the mean ± SD (*n* = 3). Independent *t*-test was performed to check difference between VFSP and VCSP (*p* < 0.05 or *p* < 0.01). All differentially expressed genes displayed significant differences between VFSP and VCSP at 0.05 confidence level.

**Figure 6 ijms-20-01012-f006:**
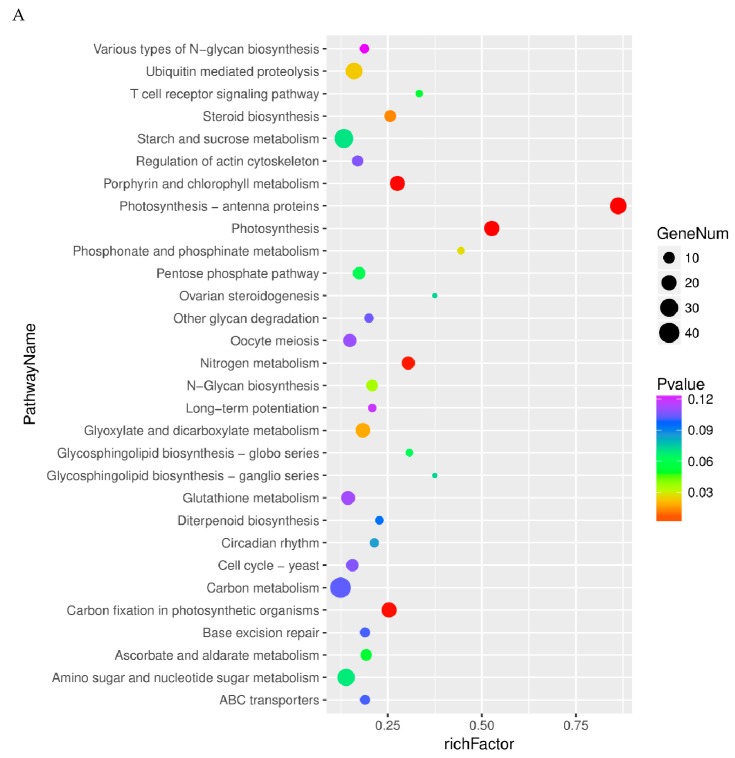
Kyoto Encyclopedia of Genes and Genomes (KEGG) Pathway enrichment analysis of DEGs identified. KEGG Pathway enrichment analysis based on the differentially up-regulated genes between VFSP and VCSP (**A**) leaves and (**B**) storage roots. KEGG Pathway enrichment analysis based on the differentially down-regulated genes between VFSP and VCSP (**C**) leaves and (**D**) storage roots.

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
