# Peer review of "Comparative Transcriptome Analysis Reveals the Transcriptional Alterations in Growth- and Development-Related Genes in Sweet Potato Plants Infected and Non-Infected by SPFMV, SPV2, and SPVG"

_ijms, 2019, doi:10.3390/ijms20051012_

Round 1

Reviewer 1 Report

The manuscript is really improved so it deserves publication.

Anyway, I suggest two minor corrections:

a)     Use more colours for histograms; Authors use mainly fuchsia, blue and green, but e.g. also red is a colour;

b)     Please indicate in Figs 1 C  and 4 D scale numbers for low value “relative expression levels”.

Author Response

Responses to reviewer 1

Dear Reviewer 1,

Thank you very much for your good questions and comments. We have looked through these questions and comments. The point to point answers are as follows:

Reviewer 1

1. Use more colours for histograms; Authors use mainly fuchsia, blue and green, but e.g. also red is a colour;

Thank you for good suggestion. Before submission of this article, all figures were edited by Spring Nature English-editing Co. Red color or others were used in the original histograms. For example, Figure 1E:

2. Please indicate in Figs 1 C and 4 D scale numbers for low value “relative expression levels”.

Thank you for good suggestion. We have tried more times to change scale numbers for low value for “relative expression levels” using Sigmaplot 10 software, but failed. It is may be related to super big value in the original histograms. 

Reviewer 2 Report

The authors report the transcriptional characterization of sweetpotato plants healthy and infected with three widespread viruses. Leaves and storage roots showed several transcriptional and metabolic alterations associated with viral infection.

The work is interesting because it improves the knowledge about the effect of viruses in sweetpotato. However, the authors do not take full advantage of the data produced, and I suggest some changes that could be useful to improve the manuscript.

·     Please deposit the raw sequencing data of RNA-seq in a public database.

·     In the Table S5, add the functional annotation available of the sweetpotato genes.

·     Please report the Fold changes for all genes.

·     The 8 tables included in the text are boring and with low impact on the manuscript. Since the essential data are already describe in text, I suggest moving these table in the supplementary materials and adding some figures obtained by Bingo plugin which can have a greater impact on the reader.

·     Please, rewrite the results and the discussion to create some links between the phenotypes (Figs 1 and 2) of plants and the transcriptomic data. There is a lot of potential to produce a very interesting discussion if the authors link the contents of chlorophyll, starch, vitamin C, etc with the transcriptomic data. Please avoid just a boring list of GO categories.

·     Please also discuss the large differences in DEGs numbers and in GO terms observed between leaf and storage roots.

·     In the abstract, please report the full names of the viruses and reduce the text, in my opinion the abstract is too long.

·     Fig.4: please add the statistical analysis and report in the caption the functional annotation of each gene.

Author Response

Responses to reviewer 2

Dear Reviewer 2,

Thank you very much for your good questions and comments. We have looked through these questions and comments. The point to point answers are as follows:

Reviewer 2

1. Please deposit the raw sequencing data of RNA-seq in a public database.

Thank you for your suggestions. We are uploading the raw sequencing data of RNA-seq including 12 GTF files from sweetpotato leaf and storage root into NCBI. The number of submission is SUB5068005.

2. In the Table S5, add the functional annotation available of the sweetpotato genes. Please report the Fold changes for all genes.

We have added the functional annotation available of the sweetpotato genes and the fold changes for all genes.

3. The 8 tables included in the text are boring and with low impact on the manuscript. Since the essential data are already describe in text, I suggest moving these table in the supplementary materials and adding some figures obtained by Bingo plugin which can have a greater impact on the reader.

We have changed it.

4. Please, rewrite the results and the discussion to create some links between the phenotypes (Figs 1 and 2) of plants and the transcriptomic data. There is a lot of potential to produce a very interesting discussion if the authors link the contents of chlorophyll, starch, vitamin C, etc with the transcriptomic data. Please avoid just a boring list of GO categories.

Thank you for your good suggestions. We rewrite the results and the discussion to create some links between the phenotypes of plants and the transcriptomic data.

Please also discuss the large differences in DEGs numbers and in GO terms observed between leaf and storage roots.

We have discussed the large differences in DEGs numbers and in GO terms observed between leaf and storage roots.

In the abstract, please report the full names of the viruses and reduce the text, in my opinion the abstract is too long.

We have added the full names of the viruses, and shortened the content of abstract.

Fig.4: please add the statistical analysis and report in the caption the functional annotation of each gene.

We have added it.

Round 2

Reviewer 2 Report

The manuscript was improved after the revisions.

Some minor remarks:

Please report the NCBI submission number in the manuscript.

why in Fig 5 there are several KEGG pathways referring to metabolisms of human/animal (tuberculosis, diabetes mellitus, etc) and not of plants? Please check.

Please check the numbers of supplemental data: for example, the table of primers was indicated as Table S8 in text, S6 in the caption of the table and S10 in the name of supplemental file….

In table S8 (S6? S10?), please add the functional classification of the genes near the primer pairs.

Please add statistical analysis of qRT-PCRs in Fig. 6.

Author Response

Responses to reviewers

Dear Reviewers,

Thank you very much for your good questions and comments. We have looked through these questions and comments. The point to point answers are as follows:

1. Please report the NCBI submission number in the manuscript.

We have added it in the part of materials and methods in the manuscript.

2. Why in Fig 5 there are several KEGG pathways referring to metabolisms of human/animal (tuberculosis, diabetes mellitus, etc) and not of plants? Please check.

 Because all species are selected when searching for a database, there are specific KEGG pathways for human, animal or microorganism to occur. We have removed these pathways.

3. Please check the numbers of supplemental data: for example, the table of primers was indicated as Table S8 in text, S6 in the caption of the table and S10 in the name of supplemental file….

We have corrected these errors.

4. In table S10, please add the functional classification of the genes near the primer pairs.

We have added the functional classification of 26 genes.

5. Please add statistical analysis of qRT-PCRs in Fig. 6.

We have added statistical analysis of qRT-PCRs in Fig. 6.

This manuscript is a resubmission of an earlier submission. The following is a list of the peer review reports and author responses from that submission.

Round 1

Reviewer 1 Report

The manuscript entitled “Comparative transcriptome analysis reveals the transcriptional alterations in growth- and development-related genes in virus-free and virus-carrying sweet potato” is a well written paper about gene regulation in virus-infected plants. However, some major modifications are requested before publishing. In particular, details about VCSP and VFSP in relation to pathogens retrieved and diagnostic tests need to be better defined. Transcriptome analysis must be referred to a specific (and detailed) health status, in which viruses present in VCSP/VSFP are to be known (or their absence has to be certain). In this MS, the classification of plants within the two classes seems to be based on symptoms, but this approach is inappropriate with regard to a transcriptome analysis. Moreover, it is not clear: 1) how virus-free plant are assessed; 2) the natural or artificial condition of infected seedling/plants; 3) which viruses (of the three cited only in Results) belong to infected plants (all of them?); 4) which viruses are checked (just the three cited in Results?)

Abstract

Line 22. VCSP is not an informative definition, because transcriptome analysis must be related to specific virus infection(s). Please details which viruses are involved. Furthermore, the Authors’ definition of VCSP should lead to consider the opposite thesis as virus-free (as reported by Authors indeed). However, how can Authors be sure that the plants are really virus-free? Some test such as dsRNA should be carried out in advance to define a plant as totally virus-free. Otherwise, if some viruses are checked (i.e. those related to sweet potato virus disease, or those cited in 97-98), the right definition is virus-checked plants. But, in this latter case, the definition of VCSP is wrong, because infected plants should be referred to specific infection (those who are absent in virus-checked plants).

Introduction

Line 54. Please change in “viruses”

Line 58. Please convert value to a reference value such as US dollar (or add money change).

Lines 61-62. Please better define virus names (i.e. Sweet potato virus G). Furthermore, first letter of virus name should be capital (i.e. Potato virus Y)

Lines 62-63. Author should pay attention to the use of italic in order to avoid confusion between virus or disease. Besides use of italic for virus name are not mandatory, I found its use appropriate (as reported in this paper). However, if italic is used for virus, should be avoided for diseases. Thus, sweet potato virus disease should not be reported in italic.

Line 67. Please define previously uncited acronyms (SPCSV, SPMMV).

Line 88. It is the first time that VFSP and VCSP, so Authors must explain these acronyms. However, I found the VCSP definition too generic, because it can be related to plants infected by one virus as well as by tens. Furthermore, each virus can potentially interact differently with plant, leading to very different transcriptome profile. Later, in Result section, 3 viruses were found in VCSP but it is not clear if 1) all plants are infected by all of them; 2) other viruses are checked.

Results

Lines 96-97. I do not find this experimental trial in M&M.

Discussion

Lines 290-291. Here, the reader should understand that all diseased plants are infected by (only?) three viruses. This consideration should be clearly reported since declaration of aims.

M&M

Lines 337-338. This sentence is central for the manuscript acceptance. Which viruses are checked for both plant groups? Are viruses artificially inoculated? Are each VCSP plant infected by the same virus(es)? How can you exclude the presence of viruses in VFSP seedling? How do you evaluate health status of seedling?

Figures

The size of Figure 4 should be increased.

Reviewer 2 Report

The manuscript tried to find the transcriptional differences between VFSP and VCSP in sweet potato. Topic is interesting and fit for the aims and scope for Molecular Sciences. The work appears to have been carefully done. However, the manuscript contains some serious weakness which need to be revised.

Suggestions for revision as well as some comments are the following:

- In this study, the authors conducted RNA-seq analysis. However, the results obtained from the experiment were easily expected from the change of appearance (e.g. leaf color change). The topic of this work is interesting, however, in my opinion, considering the quality of Molecular Sciences and its readers interest, more data such as small RNA-seq must be required since the immune response seems to be a key as they mentioned in introduction (L76-84).

-Figure 1D: The dataset of VFSP must be in right side as the other panels.

-L122: The summary for RNA-seq analysis must be added.

-L130-262: These parts are less informative. These parts must be shortened.

-L263-279: Detailed information about the genes must be presented as a table.

-Discussion is inadequate. The authors should discuss about the mechanisms underlying the disorders using their data and references.

-Other minor points are not pointed out at this time, but there are some points to be revised. The authors should carefully check the manuscript accordingly.